# Electron Correlations in Sequential Two-Photon Double Ionization of an Ar Atom

Boris M. Lagutin [1], Ivan D. Petrov [1], Victor L. Sukhorukov [2,3,*], Victor A. Kilin [4], Nikolay M. Novikovskiy [2,3], Philipp V. Demekhin [3] and Arno Ehresmann [3]

1   Rostov State Transport University, 344038 Rostov-on-Don, Russia
2   Institute of Physics, Southern Federal University, 344090 Rostov-on-Don, Russia
3   Institute of Physics, Center for Interdisciplinary Nanostructure Science and Technology (CINSaT), University of Kassel, 34132 Kassel, Germany
4   Tomsk Polytechnic University, 634050 Tomsk, Russia
*   Correspondence: vlsu16@mail.ru

**Abstract:** Sequential two-photon ionization is a process that is experimentally accessible due to the use of new free-electron laser sources for excitation. For the prototypical rare Ar gas atoms, a photoelectron spectrum (PES) corresponding to the second step of the sequential two-photon double ionization (2PDI$^{II}$) at a photon excitation energy of 65.3 eV was studied theoretically with a focus on the consequences of electron correlations in the considered process. The calculation predicts many intense lines at low photoelectron energies, which cannot be explained on the basis of a one-electron approximation. The processes that lead to the appearance of these lines include many-electron correlations, either in the first or second step of photoionization. A significant fraction of the intensity of the low-energy part of PES is associated with the Auger decay of the excited states formed at the second step of 2PDI. The shape of the low-energy part of the 2PDI$^{II}$ PES is expected to be dependent on both the energy of photon excitations and the flux of the exciting beam.

**Keywords:** sequential photoionization; double ionization; photoelectron spectra; argon; many-electron correlations; free-electron laser

## 1. Introduction

The first studies of multiphoton photoionization of atoms by free electron lasers (FELs) focused on the charge distributions of ions (see [1–3] and a brief review [4]). Subsequently, it became possible to study photoelectron spectra (PES) due to the multiphoton ionization of atoms with energy and angular resolution [5,6] (see also the review devoted to studies of double- and triple-sequential ionization of atoms [7]).

If the photon excitation energy of an intense photon beam interacting with individual atoms exceeds the ionization threshold of an atom, then the most probable process among the different mechanisms of multiphoton ionization is sequential photoionization [6–8]. The theory to describe sequential two-photon double ionization (2PDI) [9,10] has, in the past, been applied to an experiment performed for Ar when the impinging-photon energy was less than the $3s$ ionization threshold [4,11]. In this case, the interpretation of the experimental data can be limited to considering the main $3p \dashrightarrow \varepsilon\ell$ channel only, i.e., considering the variety of $3s^23p^6 \dashrightarrow 3s^23p^4\varepsilon\ell\varepsilon'\ell'$ channels. At larger photon excitation energies, $\omega$, e.g., when $\omega$ exceeds the $3s$ ionization threshold, it is expected that the influences of many electron correlations will dramatically increase, considerably complicating the interpretation of an experiment. This conjecture is based on past findings, where considerable correlative contributions were identified for conventional single-photon ionization of Ar at these photon excitation energies, namely inter-shell correlations [12–15] and the dipole polarization of electron shells [16].

The main goal of the present paper is, therefore, to study the influences of the most significant single-electron and double-electron excitations on the 2PDI of Ar. We calculated the 2PDI PES of Ar at a photon excitation energy of $\omega = 65.3$ eV for which an experiment was planned in the near future. We limited ourselves to computing the 2PDI$^{\text{II}}$ PES obtained for the second step of the sequential two-photon double ionization, which can be extracted from the total signal using its quadratic flux dependence [17].

## 2. Two-Photon Processes Involving Valence and Sub-Valence Electrons

Possible processes resulting in lines in the 2PDI$^{\text{II}}$ PES of Ar predicted for a photon excitation energy of 65.3 eV are presented in Figures 1 and 2.

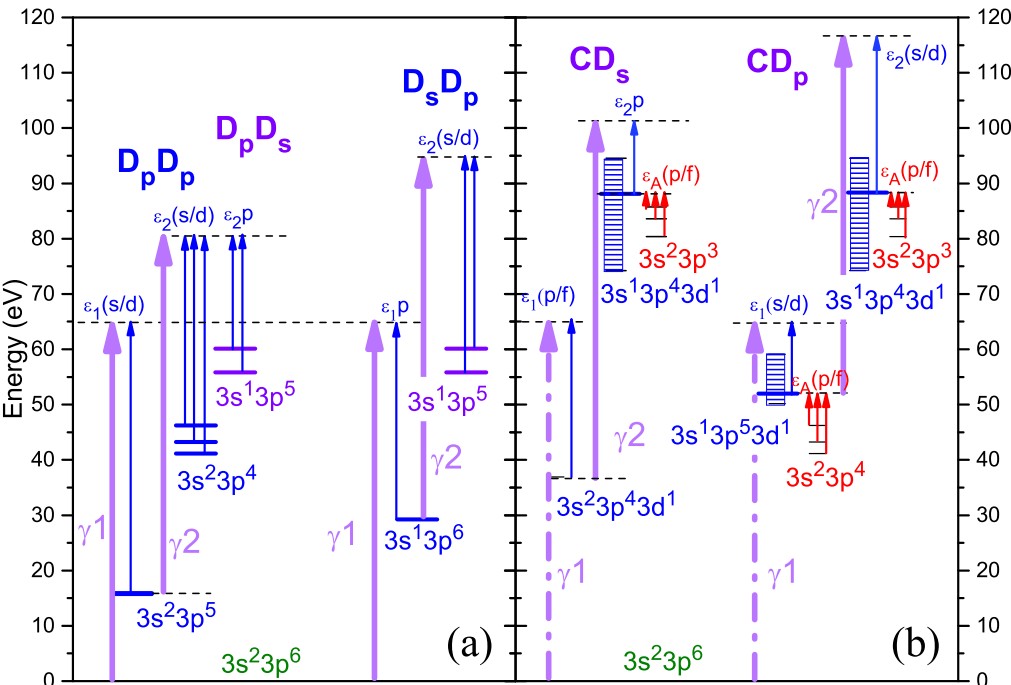

**Figure 1.** Energy scheme of the sequential two-photon double ionization of argon. (**a**) Direct one-electron ionization at each step. (**b**) Correlational processes relevant to the first ionization step. Designations: bold violet arrows—energies of the exciting photons: solid arrows—direct (*D*) photoionization processes and dash-dotted arrows—correlational (*C*) processes in the photoionization. Thin arrows—kinetic energy of the emitted photoelectrons (blue) and Auger electrons (red). Vertical rectangles represent a set of energy levels of the proper configuration allowed according to the selection rules. Energy levels were computed in the HF approach with DPES. The heights of the rectangles were estimated using the "transition array" technique [18,19].

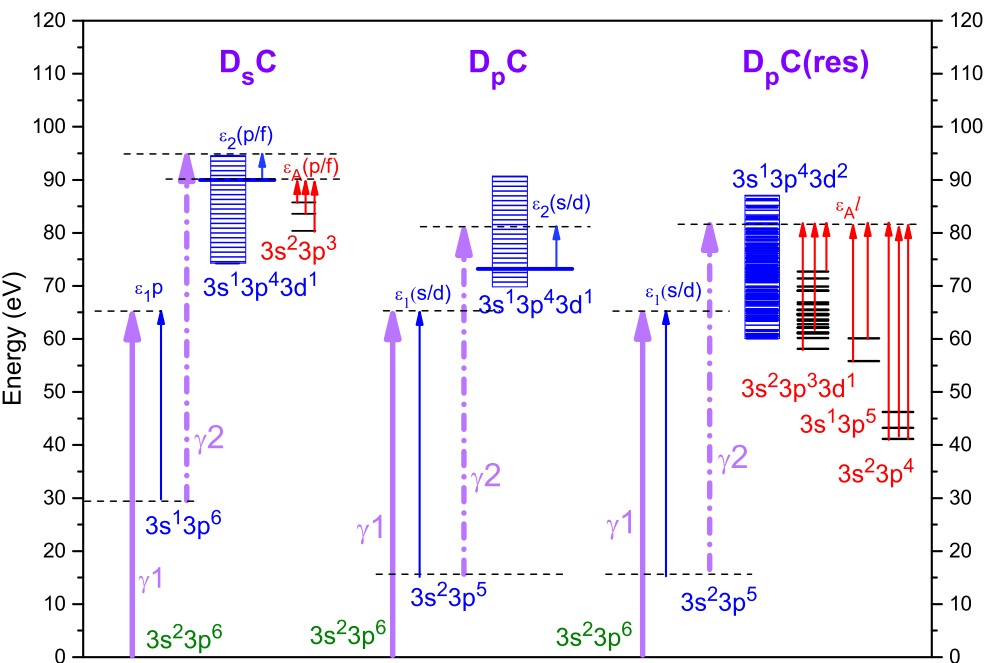

**Figure 2.** Energy schemes of the sequential two-photon double ionization of argon with a correlational process at the second ionization step. Designations are the same as in Figure 1.

### 2.1. Direct Processes

In the following, we will call 2PDI$^{\text{II}}$ processes, where one electron from a certain shell is removed by a first photon and a second electron is removed from the shell of the ion by a second photon; "direct" processes are abbreviated by the symbol D in the depicted graphs; 2PDI$^{\text{II}}$ process, removing two $3p$ electrons from Ar ($D_p D_p$), is shown in panel (a), column $D_p D_p$:

$$3s^2 3p^6 + \gamma_1 \dashrightarrow 3s^2 3p^5 + \varepsilon_1 \ell (\ell = 0, 2); \tag{1}$$

$$3s^2 3p^5 + \gamma_2 \dashrightarrow 3s^2 3p^4 + \varepsilon_2 \ell (\ell = 0, 2). \tag{2}$$

This process was studied in detail both experimentally and theoretically [4,7,11,17,20,21]. It results in three 2PDI$^{\text{II}}$ PES lines with the kinetic energy of electrons in the region from 33 to 40 eV. This part of the spectrum is the most intense.

In column $D_p D_s$ of Figure 1a another two-step process is shown in which the first step is as process (1) above, but where in the second step, a $3s$ electron is removed from the ion

$$3s^2 3p^5 + \gamma_2 \dashrightarrow 3s^1 3p^5 + \varepsilon_2 p \tag{3}$$

In the final stage of the process (3), the strong dipole polarization of the electron shells (DPES) [16] was taken into account. This effect is described by $3p3p - 3sn\ell(\ell = 0, 2)$ excitations in the current approach. As a result, the two lowest levels shown in column $D_p D_s$ of Figure 1a mainly contains the $3s^1 3p^5 (^3P)$ and $3s^1 3p^5 (^1P)$ basis states. Higher levels (not shown in Figure 1a) mainly contain $3s^2 3p^3 n\ell(\ell = 0, 2)(LS)$ basis states and result in low-energy satellite lines (sat).

In the 2PDI$^{\text{II}}$ process, a $3s$ electron is removed from the $3s$ shell of the atom, and in the second step, a $3p$ electron is removed from the singly charged ion

$$3s^2 3p^6 + \gamma_1 \dashrightarrow 3s^1 3p^6 + \varepsilon_1 p; \tag{4}$$

$$3s^1 3p^6 + \gamma_2 \dashrightarrow 3s^1 3p^5 + \varepsilon_2 \ell (\ell = 0, 2); \tag{5}$$

is depicted in column $D_s D_p$ of Figure 1a. Processes (4) and (5) result in photoelectron lines for electrons with kinetic energies larger than 30 eV.

### 2.2. Correlational Processes

The direct ($D$) processes considered above are transitions that are described by one-electron electric dipole interaction. The other atomic electrons are spectators in the transitions. Below, we consider correlational ($C$) processes influencing both the first and second steps of the sequential two-photon double ionization.

The $D_sC$ column of Figure 2 (left column) shows a two-step process that leads to the appearance of 2ph(II) lines with electron energy in the range from 0 to 20 eV. The first step of this process is the direct ($D$) transition (4). Note that in this transition, the important inter-shell correlation between the $3s^13p^6\varepsilon_1p$ and $3s^13p^5\varepsilon_2d$ channels [14] in the final state was taken into account. The second step is possible only due to electron correlations ($C$) either in the initial or the final state, since the direct single-electron transition

$$3s^13p^6 + \gamma_2 \dashrightarrow 3s^13p^43d^1 + \varepsilon_2\ell(\ell = 1,3) \tag{6}$$

is forbidden due to the selection rule.

Possible correlational transitions of the second step of the two-photon $D_sC$ processes in the lowest order of the Coulomb interaction are the following:

$$3s^13p^6 \longrightarrow 3s^13p^43d^1\varepsilon'\ell'(\ell' = 0,2,4) \dashrightarrow 3s^13p^43d^1\varepsilon_2\ell(\ell = 1,3) \tag{7}$$

$$3s^13p^6 \longrightarrow 3s^23p^43d^1 \dashrightarrow 3s^13p^43d^1\varepsilon_2p \tag{8}$$

$$3s^13p^6 \longrightarrow 3s^13p^4\varepsilon'\ell'(\ell' = 1,3)\varepsilon_2\ell \dashrightarrow 3s^13p^43d^1\varepsilon_2\ell(\ell = 1,3) \tag{9}$$

$$3s^13p^6 \longrightarrow 3s^03p^53d^1\varepsilon_2\ell(\ell = 1,3) \dashrightarrow 3s^13p^43d^1\varepsilon_2\ell(\ell = 1,3) \tag{10}$$

$$3s^13p^6 \dashrightarrow 3s^13p^5\varepsilon'\ell'(\ell' = 0,2) \longrightarrow 3s^13p^43d^1\varepsilon_2\ell(\ell = 1,3) \tag{11}$$

$$3s^13p^6 \dashrightarrow 3s^03p^6\varepsilon_2p \longrightarrow 3s^13p^43d^1\varepsilon_2p \tag{12}$$

$$3s^13p^6 \dashrightarrow 3s^13p^53d^1 \longrightarrow 3s^13p^43d^1\varepsilon_2\ell(\ell = 1,3) \tag{13}$$

In all Equations (7)–(13), and following, the dashed arrows denote the electric dipole interaction and the solid arrows denote the Coulomb interaction.

We estimated the amplitudes (7)–(13) using the "transition array" technique [18,19], which was created to solve astrophysical problems in order to avoid cumbersome "level-to-level" calculations in atoms with a large number of open shells. In the case of the problem solved in the present paper, applying the "transition array" technique required calculating discrete and continuum atomic orbitals (AOs) for channels with orbital numbers $\ell = 0 - 4$, in addition to the core AOs. Discrete AOs with an average radius of up to 20 a.u. and continuum AOs with energy $\varepsilon = 800$ Ry were computed in the frozen Hartree–Fock (HF) field of $Ar^{2+}$. In this configuration space, all single- and double-electron excitations in the initial and final states were taken into account, contributing to the 2PDI process under consideration. Among hundreds of channels contributing to the 2PDI (excluding the direct channels $D_pD_p$ and $D_pD_s$) the greatest contributions to the 2PDI were given by those that contained a $3d$ electron (see Equations (7)–(13)). At the same time, among all correlational amplitudes (7)–(13), amplitude (7) had the largest value. The main aim of the present work was to identify the main processes contributing to the low-energy part of the photoelectron spectrum rather than a detailed quantitative description of the spectral structure. Therefore, in computing the photoionization cross section (PICS) of the second step of the $D_sC$ process, we restricted our description to the most significant amplitude (7) only and neglected the DPES.

The final states stemming after the $D_sC$ two-photon double-ionization, decay by the Auger process

$$3s^13p^43d^1 \longrightarrow 3s^23p^3\varepsilon_A\ell(\ell = 1,3) \tag{14}$$

The Auger electrons of process (14) also contributed to the 2PDI$^{II}$ spectrum at low kinetic energies. This is depicted in column $D_sC$ of Figure 2 by red vertical arrows.

Column $D_pC$ of Figure 2 represents a correlational process similar to $D_sC$, but creates the $3s^23p^5$ ionic state after the first ionization step.

The $D_pC(res)$ correlational process depicted in the right column of Figure 2 is of particular interest. In this process, the second photon does not knock out the electron, but excites it into states of the $3s^13p^43d^2$ configuration, with a subsequent Auger decay. Since the excitation to the $3s^13p^43d^2$ states is resonant, the shape of the 2PDI$^{II}$ spectrum would strongly depend on the photon excitation energy. The "Auger part" of the photoelectron spectrum should be rich due to a large number of possible Auger decays (see multiple red arrows in the right column of Figure 2):

$$3s^13p^43d^2 \longrightarrow 3s^13p^5\varepsilon_A\ell(\ell = 1,3) \tag{15}$$

$$3s^13p^43d^2 \longrightarrow 3s^23p^4\varepsilon_A\ell(\ell = 0,2,4) \tag{16}$$

$$3s^13p^43d^2 \longrightarrow 3s^23p^33d^1\varepsilon_A\ell(\ell = 1,3) \tag{17}$$

Correlational transitions may also take place in the first step of 2PDI. The $CD_s$ column of Figure 1b shows a two-step process, the first step of which is possible only because of the configuration interaction in the initial or the final state, since the direct transition

$$3s^23p^6 + \gamma_1 \dashrightarrow 3s^23p^43d^1 + \varepsilon_1\ell(\ell = 1,3) \tag{18}$$

is dipole forbidden. One of the possible correlation transitions leading to the appearance of the $3s^23p^43d^1$ ionic state after the first ionization step is the following:

$$3s^23p^6 \dashrightarrow 3s^23p^5\varepsilon'\ell'(\ell' = 0,2) \longrightarrow 3s^23p^43d^1\varepsilon_1\ell(\ell = 1,3). \tag{19}$$

The second step of the $CD_s$ process is a direct transition

$$3s^23p^43d^1 + \gamma_2 \dashrightarrow 3s^13p^43d^1 + \varepsilon_2 p. \tag{20}$$

The process $CD_s$ result in lines in the same region of the 2PDI$^{II}$ PES as the $D_sD_p$ process, i.e., with an electron kinetic energy larger than 30 eV.

Another possible two-photon process is depicted in the $CD_p$ column of Figure 1b. Similar to the $CD_s$ case, the first step is possible due to the configuration interaction either in the initial or final configuration only, since the direct transition

$$3s^23p^6 + \gamma_1 \dashrightarrow 3s^13p^53d^1 + \varepsilon_1\ell(\ell = 0,2) \tag{21}$$

is dipole forbidden. One of the possible correlational transitions leading to the appearance of the $3s^13p^53d^1$ ionic state after the first ionization step is the following:

$$3s^23p^6 \dashrightarrow 3s^23p^5\varepsilon'\ell'(\ell' = 0,2) \longrightarrow 3s^13p^53d^1\varepsilon_1\ell(\ell = 0,2). \tag{22}$$

The second step of the $CD_p$ process is the direct transition

$$3s^13p^53d^1 + \gamma_2 \dashrightarrow 3s^13p^43d^1 + \varepsilon_2\ell(\ell = 0,2). \tag{23}$$

The peculiarity of the $CD_p$ process is the fast Auger decay of the intermediate $3s^13p^53d^1$ states

$$3s^13p^53d^1 \longrightarrow 3s^23p^4\varepsilon_A\ell(\ell = 1,3) \tag{24}$$

which reduce the intensities of respective 2PDI$^{II}$ lines (if the intensity of the exciting radiation is small). A similar situation was considered in [22], where the process of the two-photon ionization of the Ne atom was studied.

Our estimates show that the average Auger width of the $3s^13p^53d^1$ states is about 1 eV, whereas the largest photoionization cross section of these states at the energy of 65.3 eV does not exceed 0.5 Mb. Then, at the exciting–radiation flux of $1.5 \times 10^{15}$ W/cm$^2$ [4], the probability of process (23) is less than $10^{-9}$. When the probability of process (23) reaches a noticeable value at a radiation flux of more than $10^{16}$ W/cm$^2$, it will become 5%.

Thereby, at a lower flux of the exciting radiation, the Auger decay (24) will most likely occur after the first ionization step in the $CD_p$ process. Consequently, the ionization of the state of a doubly charged ion will occur in the second step as

$$3s^23p^4 + \gamma_2 \dashrightarrow 3s^23p^3 + \varepsilon_2\ell(\ell = 0, 2). \tag{25}$$

or

$$3s^23p^4 + \gamma_2 \dashrightarrow 3s^13p^4 + \varepsilon_2 p. \tag{26}$$

The kinetic energies of electrons in process (25) exceed 20 eV, and process (26), similar to process (3), with low intensity. Therefore, the $CD_p$ process will not be discussed later.

## 3. Calculation Details

The atomic orbitals (AOs) used in the calculation of the cross sections for the first ionization step were obtained by solving the non-relativistic Hartree–Fock (HF) equations for the ground $1s^22s^22p^63s^23p^6$ configuration of Ar. The AOs of continuum electrons were calculated by solving the HF term-dependent equations for the $3s^23p^5\varepsilon s(^1P)$, $3s^23p^5\varepsilon d(^1P)$ configurations in the case of the $D_pD_p$ process and for the $3s^13p^6\varepsilon p(^1P)$ configuration in the case of the $D_sC$ process with frozen core AOs.

The total experimental cross sections of Ar [23] were compared with the presently computed $\sigma_{3p}(\omega)$ in Figure 3. One can recognize that the HF cross sections calculated in the length form (dash curve) and velocity form (short dash curve) of the dipole transition operator differed by more than 1.5 times at the threshold.

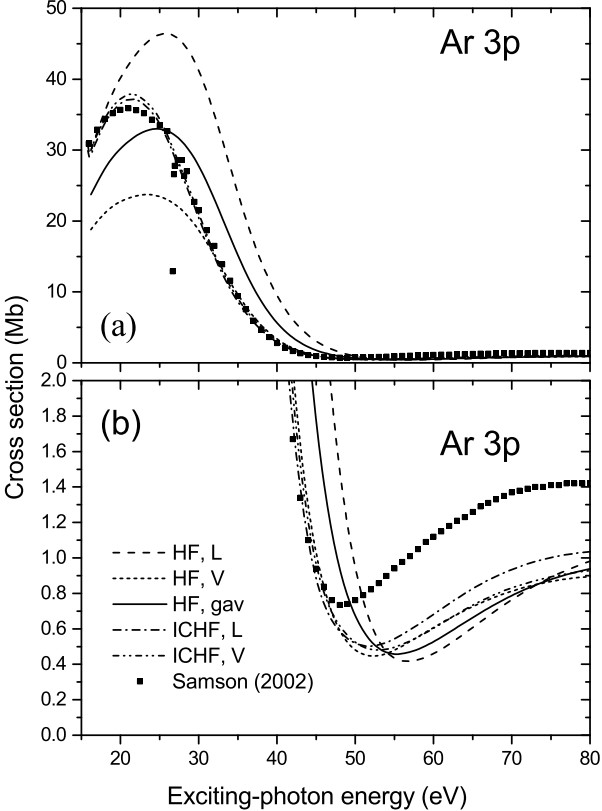

**Figure 3.** Total experimental PICS of Ar [23] (Samson 2002) and presently computed PICS of the Ar 3p shell in the single-electron HF approximation in length $\left(\sigma_L^{HF}\right)$ and velocity $\left(\sigma_V^{HF}\right)$ forms and the geometric average between the length and velocity forms $\left(\sigma_{gav}\right)$. Results of the calculation, taking into account intra-shell correlation(s) (ICHF) in length and velocity forms are depicted as well. Panel (**b**) represents an enlarged part of panel (**a**) in the region of the Cooper minimum.

We also calculated the $3p$ shell PICS, taking into account the intra-shell correlation(s) (ICHF) via a technique described in [14]. These cross sections (dash–dot curve for the length form and dash–dot-dot curve for the velocity form) are very close to each other and to the experimental cross sections in the photon energy range from the threshold up to 30 eV. At higher photon energies, of more than 30 eV, the theoretical PICS of the $3p$ shell of Ar differ from the experimental ones (see panel (b)), since for these energies, in addition to photoionization of the $3p$ shell, photoionization of the $3s$ shell is possible to the main $3s^1 3p^6$ and satellite $3s^2 3p^4 n\ell$ ionic states.

Intensities of the photoelectron lines $I_{ij}(\omega)$ were calculated using the formula

$$I_{ij}(\omega) \sim \sigma_{1i}(\omega) \cdot \sigma_{2ij}(\omega), \tag{27}$$

where $\sigma_{1i}(\omega)$ is the cross section to the first 2PDI step—photoionization of the neutral atom leading to the formation of a single-charged ion state $i$; $\sigma_{2ij}(\omega)$ is the cross section for the second 2PDI step—photoionization of the ionic state $i$ leading to the formation of a double-charged ion state $j$.

Cross sections $\sigma_{1i}(\omega)$ and $\sigma_{2ij}(\omega)$, calculated within the Hartree–Fock (HF) approximation in the length ($\sigma_L^{HF}$) and velocity ($\sigma_V^{HF}$) gauges could have significant differences that disappear when taking into account the inter-shell and intra-shell correlations, applying all orders of the perturbation theory [14,15]. In some cases, cumbersome calculations associated with taking into account these correlations can be avoided if the geometric average (gav) value of these cross sections, $\sigma_{gav}$, is used:

$$\sigma_{gav} = \left( \sigma_L^{HF} \cdot \sigma_V^{HF} \right)^{1/2} \tag{28}$$

Figure 3 illustrates that, in the case of Ar $3p$-photoionization, using $\sigma_{gav}$ achieves an inaccuracy of 10% when computing PICS in the photon excitation energy range of interest. Therefore, the $\sigma_{gav}$- approximation was used throughout the present paper.

The following set of AOs was used in our calculations of the second step of 2PDI$^{II}$. In the $D_p D_p$ and the $D_p D_s$ processes, the AOs were obtained by solving the HF equations for the $1s^2 2s^2 2p^6 3s^2 3p^5$ configuration. The AOs of continuum electrons for the $D_p D_p$ process were calculated by solving the term-depending HF equation for the $3s^2 3p^4 (^1S, \,^1D, \,^3P) \varepsilon\ell(LS)$ states with frozen core AOs. The AOs of continuum electrons for the $D_p D_s$ process were calculated similarly but for the $3s^1 3p^5 (^1P, \,^3P) \varepsilon p(LS)$ states using frozen core AOs. The total cross section was obtained as a sum of partial cross sections over the $LS$ quantum numbers.

In order to take into account the DPES in the second step of the $D_p D_s$ process, the interaction between the $3s^1 3p^5$ and $3s^2 3p^3 nd$ ($n = 3, 4$) configurations was taken into account.

The cross sections for the second ionization step in the case of the $D_s C$ process were computed applying the amplitude (7) only, which is described in more detail. The expression for the photoionization cross section is

$$\sigma_2 \left( 3s^1 3p^6 \longrightarrow 3s^1 3p^4 3d^1, \omega \right) = \frac{4}{3} \pi^2 \alpha a_0^2 \omega^{\pm 1} \sum_\ell \left| A \left( 3s^1 3p^6 \longrightarrow 3s^1 3p^4 3d^1 \varepsilon_2 \ell \right) \right|^2, \tag{29}$$

where the signs ($+$) and ($-$) correspond to the length and velocity gauges of the transition dipole operator $D$, respectively; $\omega$ determined by $E(3s^1 3p^6) + \omega = E(3s^1 3p^4 3d^1) + \varepsilon_2$ stands for the photon excitation energy in atomic units; $\alpha = 1/137.036$ is the fine-structure constant; and the square of the Bohr radius $a_0^2 = 28.0028$ Mb converts the atomic units to cross sections in $Mb = 10^{-22}$ m$^2$.

The transition amplitude (7) is:

$$A\left(3s^13p^6 \longrightarrow 3s^13p^43d^1\varepsilon_2\ell\right) =$$
$$\sum_{\varepsilon'\ell'} \frac{\langle 3s^13p^43d^1\varepsilon_2\ell|D|3s^13p^43d^1\varepsilon'\ell'\rangle\langle 3s^13p^43d^1\varepsilon'\ell'|H^{ee}|3s^13p^6\rangle}{-IP(3s^13p^43d^1) - \varepsilon'} \quad (30)$$

where $D$ is the dipole transition operator; $H^{ee}$ is the Coulomb interaction operator; $IP(3s^13p^43d^1)$ is the ionization potential for the configuration $3s^13p^43d^1$ in respect with the $3s^13p^6$ state; the summation/integration over $\varepsilon'$ includes both the discrete and continuum states.

The calculation was performed using the AOs obtained by solving the HF equations for the $1s^22s^22p^63s^13p^6$ configuration. The $3d$ AO was obtained in the $1s^22s^22p^63s^13p^43d^1$ configuration. The continuum AOs for the intermediate and final $3s^13p^43d^1\varepsilon'\ell'$ and $3s^13p^43d^1\varepsilon_2\ell$ configurations were calculated using frozen core AOs averaged over the configuration. Since the amplitude (7) contained divergent radial integrals $\langle\varepsilon_2\ell|r|\varepsilon'\ell'\rangle$, it was calculated using the correlation function method [24].

When calculating the energy levels of the $1s^22s^22p^63s^13p^43d^1$ configuration, the AOs obtained by solving the HF equations for this configuration were used. The same atomic orbitals were used to calculate the Auger decay probabilities (14).

Intensities of the Auger components for the PES were calculated using the formula

$$I(ij \longrightarrow LS, \omega) = I_{ij}(\omega) \cdot \varkappa(j \longrightarrow LS), \quad (31)$$

where $I(ij \longrightarrow LS, \omega)$ is the intensity of the Auger line corresponding to the decay of the $3s^13p^43d^1(j)$ state to the configuration $3s^23p^3(LS)\varepsilon\ell$; $I_{ij}(\omega)$ is the intensity (27) of the photoelectron line and $\varkappa(j \longrightarrow LS)$ is the Auger yield computed as

$$\varkappa(j \longrightarrow LS) = \Gamma(j \longrightarrow LS)\left[\sum_{L'S'}\Gamma(j \longrightarrow L'S')\right]^{-1}, \quad (32)$$

where $\Gamma(j \longrightarrow LS)$ is the partial width of the $j \longrightarrow LS$ Auger-transition and the summation in the denominator includes all terms of the configuration $3s^23p^3(LS)$, the decays that are energetically allowed. The partial Auger width in the atomic units was calculated as

$$\Gamma(j \longrightarrow LS) = 2\pi\left|\left\langle 3s^23p^3(LS)\varepsilon\ell|H^{ee}|3s^13p^43d^1(j)\right\rangle\right|^2. \quad (33)$$

## 4. Results and Discussion

The most relevant calculation of the photoelectron spectrum during the sequential two-photon double ionization of the Ar atom, known to date, was the calculation by Kiselev et al. in 2020 [11]. In that work, the photoionization cross sections (at a photon energy of 33.4 eV) were measured and computed using the R-matrix technique with the $B$-spline $R$-matrix package [25]. At this photon energy, the $D_pD_s$ direct process described in Section 2.1 and the correlational processes considered in Section 2.2 do not occur due to the insufficient energy of the incident radiation.

The experimental spectrum contains the lines of both the first and second ionization steps. In particular, the electron energies in the $3s^23p^6 + \gamma_1 \dashrightarrow 3s^13p^6\varepsilon_1p$ first-step transition are very close to those for the second-step transition $3s^23p^5 + \gamma_2 \dashrightarrow 3s^23p^4(^1D)\varepsilon_2\ell(\ell = 0,2)$. Therefore, the corresponding lines in the photoelectron spectrum overlap. As a result, it is impossible to determine the intensity of the line corresponding to the $3s^23p^4(^1D)$ state using the experimental spectrum obtained in [11].

In the present work, we calculated the 2PDI$^{II}$ PES of Ar at a photon energy of 33.4 eV. In accordance with the above, we compare the results of both calculations only. From Figure 1 of [11], we estimate that the relative line intensities in the computed 2PDI$^{II}$ PES are $I(^3P) : I(^1D) : I(^1S) = 100 : 45 : 15$. In our calculation, this ratio is 100:32:11. The two

calculations yield similar ratios, and we are, thus, confident that our calculation method provides sufficient accuracy to explain the features of a possible new experiment in the "correlation" region of exciting photon energies.

As discussed above, only transitions (1)–(4), and (7) were taken into account in the present calculations. In total, this stage required computing 3 partial PICS to the first 2DPI step (see processes (1) and (4)) and 34 PICS for the second 2DPI step (see processes (2), (3), and (7)). The PICS to the first 2DPI step (1) are depicted in Figure 3. The PICS to the first 2DPI step (4) at an exciting photon energy 65.3 eV are $\sigma_1(\omega = 65.3 \text{ eV}) = 0.22$ Mb. The largest correlational PICS of the second 2DPI step are compared with the PICS of the direct transition (2) in Figure 4. It can be seen that for all three terms $3s^2 3p^4 (^1S, {}^1D, {}^3P)$, the shapes of the PICS profiles for direct transitions are qualitatively similar to the usual $3p$ PICS (see Figure 3), which are determined by the Cooper minimum. The correlational cross section has a qualitatively different dependence on the photon excitation energy, which is mainly determined by the energy dependence of the $3d \dashrightarrow \varepsilon f$ transition and, to some extent, is a manifestation of the $3d \dashrightarrow \varepsilon f$ giant resonance. The ratio of partial cross sections for the $\varepsilon p$ and $\varepsilon f$ channels in the photon excitation energy range of 50–80 eV varied by almost three times, which suggests a significant energy dependence of the photoelectron angular distribution parameter in the specified energy range and may be of interest for experimental research.

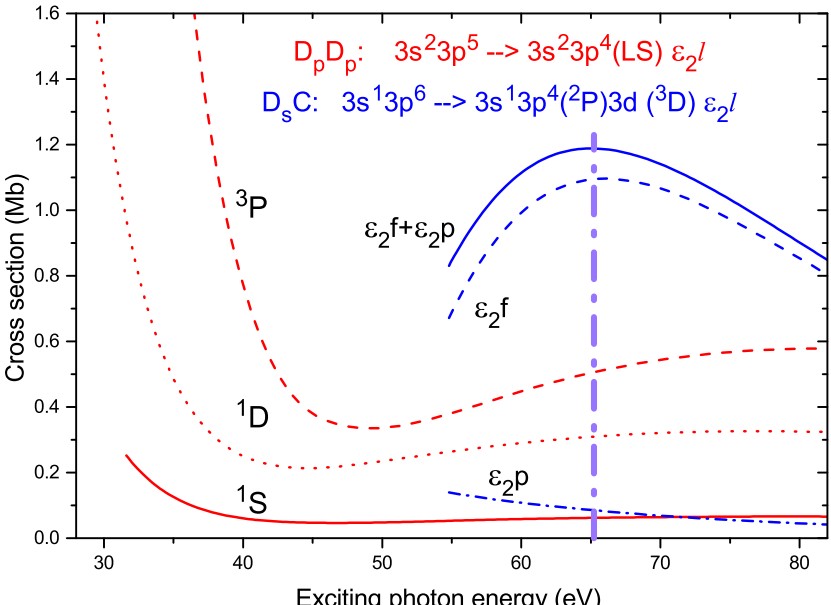

**Figure 4.** Calculated photoionization cross section(s) (PICS) for the second step of the direct $3s^2 3p^5 \dashrightarrow 3s^2 3p^4 \varepsilon_2 \ell$ and the largest correlational $3s^1 3p^6 \dashrightarrow 3s^1 3p^4 3d^1 \varepsilon_2 \ell$ 2PDI transitions of Ar. For the correlational transition, partial PICS are also shown. The vertical dash-dotted line shows the photon excitation energy used for computing the photoelectron spectrum in Figure 5.

The photoelectron spectra computed in several approximations are depicted in Figure 5. Figure 5a shows the spectrum due to direct transitions (1)–(3) only. Transitions (1) and (2) lead to the appearance of three 2PDI$^{II}$ lines in the energy range from 32 to 40 eV, corresponding to the Ar$^{2+}$ $3s^2 3p^4 (^1S, {}^1D, {}^3P)$ states, and transitions (1) and (3) lead to the appearance of a structure in the range of electron kinetic energies from 0 to 25 eV. The components of the computed spectrum were convolved using Gaussian functions, FWHM = 0.5 eV. For a better comparison, we present all of the calculated relative intensities of the groups of lines with respect to the total intensity of the $3p^4$ ($D_p D_p$) multiplet, which is considered 100%. The calculated ratio of the intensities $I(D_p D_p) : I(D_p D_s)$ is 100 : 25. Within the $D_p D_s$ structure the ratio of the $I(3s^1 3p^5)$ group to the $I(3s^2 3p^3 nd^1 (sat))$ group is 14:11.

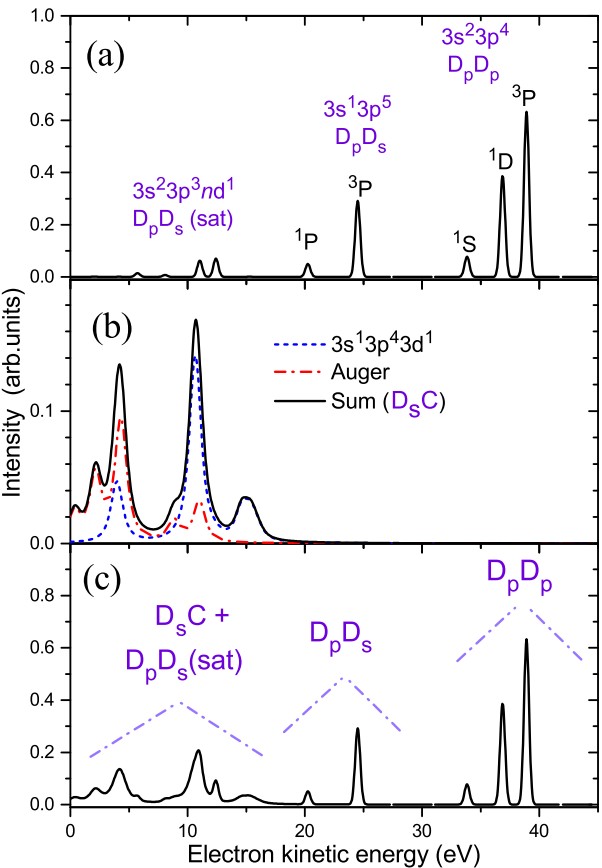

**Figure 5.** The 2PDI$^{\text{II}}$ PES of Ar for photon excitation energy 65.3 eV; (**a**) direct $D_p D_p$ and $D_p D_s$ processes; (**b**) $D_s C$ correlational process: photoelectron lines of processes (4) and (7) ($3s^1 3p^4 3d^1$); Auger lines of process (14) (Auger) and total spectrum (Sum); (**c**) the sum of the spectra depicted in panels (**a**,**b**).

The theoretical spectrum caused by the $D_s C$ process is shown in Figure 5b. Accounting for transitions (4) and (7) results in fairly intense structures in the energy range from the threshold to 20 eV (short dash curve). We also took into account the Auger decay channel (14). The lines corresponding to Auger electrons, computed according to Equations (31)–(33), are also shown in Figure 5b by the dash–dot curve. The total spectrum is presented in Figure 5b by the solid curve. For a better perception of the contributions of the photoelectron and Auger-electron parts of the spectrum, the intensity scale was increased by a factor of 5 compared to Figure 5a. Since the Auger width of the transition (14) was 1 eV on average, when constructing this part of the spectrum, its components were convolved using Lorentz functions, FWHM = 1 eV, and additionally convolved by an instrumental broadening using Gaussian functions, FWHM = 0.5 eV. Finally, the total spectrum, taking into account all processes (1)–(3), (4), (7), and (14), is shown in Figure 5c. Both correlation effects have comparable intensities with a ratio $I(3s^1 3p^4 3d^1) : I(Auger)$ equaling to 72:64 (in respect with $I(D_p D_p)$).

To summarize, the calculated in the described approximation integral intensity of PES lines for the photoelectrons with low kinetic energies, stemming from the correlational $D_s C$ process, is 136% of the $I(D_p D_p)$ intensity of the direct (non-correlational) two-photon $D_p D_p$ process associated with transitions to the Ar$^{2+}$ $3s^2 3p^4(^1S, {}^1D, {}^3P)$ states.

Further theoretical investigations could be taken into account regarding the DPES, which involves mixing the $3s^1 3p^6$ configuration with the $3s^2 3p^4 n\ell(\ell = 0, 2)$ configurations in the first ionization step (4). In the second ionization step, the amplitude (7), remaining amplitudes (8)–(13), and the DPES in these processes should be taken into account.

## 5. Conclusions

We computed a theoretical photoelectron spectrum for photoelectrons of low kinetic energies for the second step of sequential two-photon double ionization (2PDI$^{\text{II}}$ PES) of Ar at a photon excitation energy of 65.3 eV, exceeding the 3$s$ ionization threshold. The calculation takes into account the most significant electron correlations stemming from single- and double-electron excitations.

The calculation predicts that in the low-energy part of the 2PDI$^{\text{II}}$ PES, correlation satellites should be observed whose intensities exceed the intensity of the main (direct) transition to the Ar$^{2+}$ $3s^2 3p^4(^1S,\ ^1D,\ ^3P)$ states. Correlation satellites consist of lines associated with conventional photoelectron emissions and the Auger decay of the $3s^1 3p^4 n\ell^1$ states; the cross sections of both processes are comparable in magnitude. The lines of the first type should change their energy with changing photon excitation energy, while the energy of the Auger lines should remain unchanged.

The dependence of the intensities of correlation satellites may have resonant characters associated with the excitation of discrete levels at certain energies. In this case, the excitation of resonances at the second step of the two-photon process (e.g., those stemming from the second-step configuration $3s^1 3p^4 3d^2$) will contribute to the 2PDI$^{\text{II}}$ PES even at small exciting radiation fluxes. At large fluxes, there should be a contribution from resonances excited at the first step (e.g., those stemming from the first-step configuration $3s^1 3p^5 3d^1$). The estimates of the lifetime of the $3s^1 3p^5 3d^1 (LSJ)$ states showed that the contribution of these processes can be significant at fluxes exceeding $10^{16}$ W/cm$^2$.

**Author Contributions:** Conceptualization and methodology V.L.S. and A.E.; data curation, B.M.L., I.D.P., V.A.K. and N.M.N.; formal analysis, validation, visualization, and writing—original draft, B.M.L., I.D.P. and V.L.S.; funding acquisition, project administration and supervision, P.V.D. and A.E.; investigation, B.M.L., I.D.P., V.L.S., V.A.K. and N.M.N.; software, B.M.L., I.D.P., V.L.S. and V.A.K.; writing—review and editing, B.M.L., I.D.P., V.L.S., P.V.D. and A.E. All authors have read and agreed to the published version of the manuscript.

**Funding:** V.L.S., N.M.N., P.V.D., and A.E. were funded by the Deutsche Forschungsgemeinschaft (DFG)—project no. 328961117-SFB 1319 ELCH (extreme light for sensing and driving molecular chirality).

**Data Availability Statement:** All the data reported in this work are available from the correspondence author on reasonable request.

**Acknowledgments:** The authors are grateful to Markus Ilchen for drawing their attention to the problem existing in interpreting the 2PDI of Ar, for his permanent interest in the work, and for discussing the results obtained.

**Conflicts of Interest:** The authors declare no conflict of interest.

## Abbreviations

The following abbreviations are used in this manuscript:

| | |
|---|---|
| FEL | free electron laser |
| PES | photoelectron spectrum |
| 2PDI$^{\text{II}}$ | the second step of the sequential two-photon double ionization |
| FWHM | Full width on half maximum |
| PICS | photoionization cross section |
| DPES | dipole polarization of electron shells |
| HF | Hartree–Fock |

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
