# Peer review of "Electron Correlations in Sequential Two-Photon Double Ionization of an Ar Atom"

_atoms, doi:10.3390/atoms10040139_

Round 1

Reviewer 3 Report

The manuscript “Electron Correlations in Sequential Two-Photon Double Ionization of an Ar Atom” by B. M. Lagutin et al presents results for correlation satellites arising in ionic ionization, as a part of sequential two-photon double ionization. The authors shown that the satellites plat a essential role in the formation of lower energetic part of the photoelectron spectrum.

The presented results are interesting and novel, accounting the difficulties with interpretation of complex photoelectron spectrum generated in the photoionizaiotn by modern FELs facilities.

 Unfortunately the author presented too few details and shown the only one new result namely the photoelectron spectrum. In order to plot this spectrum the authors must have calculated: (1) the cross section of ionic ionization with shake-ups and (2) Auger decay partial widths for the 3s3p43d state. There are plenty of possible terms for this configuration, as well as a few terms of the residual ion.  The authors presented neither cross sections nor Auger probabilities.

 In my opinion a reader can learn nothing from the manuscript in the present form. My recommendation is to incorporate in the manuscript more details for processes (7) and (14), with clear indication of important/unimportant terms, channels and decays.

Round 2

Reviewer 3 Report

In my opinion the new version of manuscript “Electron Correlations in Sequential Two-Photon Double Ionization of an Ar Atom” by B. M. Lagutin et al has accounted the previous comments in a appropriate way and suits for publication in”Atoms” in the present form.